# Efficacy of Integrating a Novel 16-Gene Biomarker Panel and Intelligence Classifiers for Differential Diagnosis of Rheumatoid Arthritis and Osteoarthritis

**DOI:** 10.3390/jcm8010050

**Published:** 2019-01-06

**Authors:** Nguyen Phuoc Long, Seongoh Park, Nguyen Hoang Anh, Jung Eun Min, Sang Jun Yoon, Hyung Min Kim, Tran Diem Nghi, Dong Kyu Lim, Jeong Hill Park, Johan Lim, Sung Won Kwon

**Affiliations:** 1College of Pharmacy and Research Institute of Pharmaceutical Sciences, Seoul National University, Seoul 08826, Korea; phuoclong@snu.ac.kr (N.P.L.); 2018-23140@snu.ac.kr (N.H.A.); mje0107@snu.ac.kr (J.E.M.); supercanboy@snu.ac.kr (S.J.Y.); snuhmkim04@snu.ac.kr (H.M.K.); ddongq1989@snu.ac.kr (D.K.L.); hillpark@snu.ac.kr (J.H.P.); 2Department of Statistics, Seoul National University, Seoul 08826, Korea; inmybrain@snu.ac.kr (S.P.); johanlim@snu.ac.kr (J.L.); 3School of Medicine, Vietnam National University, Ho Chi Minh 700000, Vietnam; trandiemnghi@gmail.com

**Keywords:** rheumatoid arthritis, osteoarthritis, diagnostic biomarker, machine learning, meta-analysis, pathway analysis

## Abstract

Introducing novel biomarkers for accurately detecting and differentiating rheumatoid arthritis (RA) and osteoarthritis (OA) using clinical samples is essential. In the current study, we searched for a novel data-driven gene signature of synovial tissues to differentiate RA from OA patients. Fifty-three RA, 41 OA, and 25 normal microarray-based transcriptome samples were utilized. The area under the curve random forests (RF) variable importance measurement was applied to seek the most influential differential genes between RA and OA. Five algorithms including RF, k-nearest neighbors (kNN), support vector machines (SVM), naïve-Bayes, and a tree-based method were employed for the classification. We found a 16-gene signature that could effectively differentiate RA from OA, including *TMOD1*, *POP7*, *SGCA*, *KLRD1*, *ALOX5*, *RAB22A*, *ANK3*, *PTPN3*, *GZMK*, *CLU*, *GZMB*, *FBXL7*, *TNFRSF4*, *IL32*, *MXRA7*, and *CD8A*. The externally validated accuracy of the RF model was 0.96 (sensitivity = 1.00, specificity = 0.90). Likewise, the accuracy of kNN, SVM, naïve-Bayes, and decision tree was 0.96, 0.96, 0.96, and 0.91, respectively. Functional meta-analysis exhibited the differential pathological processes of RA and OA; suggested promising targets for further mechanistic and therapeutic studies. In conclusion, the proposed genetic signature combined with sophisticated classification methods may improve the diagnosis and management of RA patients.

## 1. Introduction

In accordance with the dramatically increased incidence in the older population, osteoarthritis (OA) and rheumatoid arthritis (RA) are currently among the most common causes of musculoskeletal-related chronic disability [1,2]. Depending on the case definition and joint sites under study, the prevalence of RA was at 0.5–1.1% while that of OA was much more common, ranging from 5% of the hip and 33% of the knee to 60% of the hands in adults 65 years of age or older [3,4]. RA is a chronic autoimmune disease that exhibits persistent synovial and systematic inflammation along with the existence of autoantibodies [5]. On the other hand, OA has been characterized as a non-inflammatory degenerative joint disease although synovial inflammation is a debatably important feature [6,7]. OA and RA are pathophysiologically different but share similar and overlapping features in terms of underlying mechanisms [8,9]. In addition, early-stage RA appears to be remarkably similar to other forms of arthritis, especially OA; thus, more investigations are needed to introduce better approaches for differentiating RA and OA [10].

Although laboratory tests, prominently blood test, have been believed to provide the most important information for diagnosis of arthritis, 30% of RA patients had negative rheumatoid factor or anticyclic citrullinated peptide antibodies while 40% got a normal result for erythrocyte sedimentation rate or C-reactive protein [11]. The values indicate that complementary and alternative methods are needed to increase the diagnostic accuracy of RA. Among them, synovial tissue has been used for the diagnosis and study of arthritis for decades and its substantial impact was recently reviewed [12,13]. Several studies have suggested specific immunohistochemical features of synovial tissue that separate RA (or OA) from other joint diseases or the early stage from the late stage of disease [14,15,16]. However, a recent study indicated that only a minority of patients could have a definitive diagnosis despite the high success rate of gathering synovial tissue [17]. This reduces the potential of using synovial tissue not only for abnormal detection but also for proper guidance of treatments. Moreover, investigations exploring small subsets of genes that can be used to classify RA versus OA, RA versus healthy persons and OA versus healthy persons using synovial tissues are lacking.

Herein, we searched for genome-wide transcriptional profiles from the synovial tissues of RA, OA, and normal synovial samples and looked for genetic signatures that were suitable for accurately differentiating RA from OA as well as RA from normal tissues and OA from normal tissues. The results from this investigation are expected to be helpful in some particular populations, such as elder patients with inflammatory presentation of OA and patients without typical biomarkers. In addition, we applied gene expression meta-analysis and functional analysis to explore the different biological processes involved in the pathophysiology of RA and OA. The bioinformatics-based suggestive mechanisms would be helpful for further mechanistic and therapeutic studies. In conclusion, our findings may assist the blood-based laboratory tests to improve the accuracy of diagnosis and success rate of clinical interventions.

## 2. Materials and Methods

### 2.1. Data Sets

The available Affymetrix and Illumina microarrays on gene expression in RA, OA, and healthy control (N) synovial tissues were curated from Gene Expression Omnibus with the search terms “Rheumatoid arthritis”, “Osteoarthritis”, “Synovial tissue*”, and “Synovial membrane”. As a result, the following data sets were included: GSE1919 (3 RA, 3 OA, and 5 N), GSE39340 (10 RA and 7 OA), GSE36700 (7 RA and 5 OA), GSE55457 (13 RA, 10 OA, and 10 N), GSE55584 (10 RA and 6 OA), GSE55235 (10 RA, 10 OA, and 10 N). Synovial tissues mixed from different patients in the GSE1919 data set were excluded before data processing and analysis. In total, the data sets on RA and OA included 94 patients. The comparisons between OA and normal controls and between RA and normal controls consisted of 48 and 51 samples, respectively. Information on the included data sets is presented in Appendix A. We then divided the six data sets into two groups: set A (for variable selection purpose) consisted of GSE1919, GSE39340, and GSE36700; set B (for model fitting purpose) consisted of GSE55457, GSE55584, and GSE55235. 

### 2.2. Microarray Data Preprocessing

Robust multiarray average normalization in affy package version 1.54.0 was conducted for Affymetrix-based datasets, and robust spline normalization in lumi package version 2.28.0 was applied for Illumina beadchip arrays [18,19]. The probes were mapped according to Entrez Gene ID and official gene nomenclature. Subsequently, the individual datasets were merged after removing batch effects by empirical the Bayes cross-study normalization method [20]. Following cross-study normalization, only common genes among different platforms, e.g., Affymetrix and Illumina, and datasets remained (8597 genes).

### 2.3. Variable Importance Measurement and Selection

In this study, the area under the curve permutation random forest variable importance measurement (AUC-RF VIM) was applied. AUC-RF VIM was conducted using party package version 1.2-3 [21]. Only the candidates that had a variable importance value higher than our arbitrary criterion were selected for the classification tasks.

### 2.4. Exploratory Data Analysis and Visualization

We performed the Wilcoxon rank-sum test to compare the differences of individual predictors between two groups. Additionally, principal component analysis (PCA), a data dimensionality reduction technique, was utilized to explore the underlying trends and patterns of data as well as to detect possible outliers of two comparative groups [22]. Using Metaboanalyst 3.0, a heatmap of the centered and scaled data was also applied to visualize the differences in gene expression between two groups [23]. Violin plots with overlay box plots of the predictors were constructed using violin plots with ggpubR version 0.1.5 [24]. PCA and visualization were performed using FactoMineR version 1.35 and factoextra version 1.0.4 [25,26].

### 2.5. Supervised Machine Learning

RA and OA were classified using classification and regression training (caret) package version 6.0.77 [27]. Random forests (RF), an aggregation or ensemble of numerous decision trees, each of which utilizes resampled data and branches out based on a randomly chosen subset of features, was applied as the main classifier in the current study [28]. Four other commonly used machine learning classification techniques that belong to different categories of algorithms were also employed: k-nearest neighbors (kNN), naïve Bayes, support vector machines with polynomial kernel (SVM), and a tree-based method (we call it C5.0, named after the R function). Each classifier has one or several tuning parameters that need to be optimized. Grid search or random search with the tune length of 10 were applied when applicable [29]. First, we set the number of trees consisting of random forests to 500, which is known to suffice and thus is commonly used in practice. Additionally, the number of randomly chosen variables at each split was tuned from one to 16 in RF. The naïve Bayes classifier was optimized with a Gaussian kernel of data-driven bandwidth. The number of closest training examples (k) in kNN was tuned to define the value of k that resulted in optimal performance of the model. It is well known that the SVM classifier can be found by solving a quadratic programming via the dual of the primal problem where the cost constant C>0 penalizes the margins. In our analysis, the SVM classifier relied on the polynomial kernel represented by
K(x,y)=γ(xTy+1)d, x,y ∈ℝp
where γ>0, d≥1 are the scale parameter and degree of the polynomial, respectively. The tuning parameters C, γ,d were tuned using the random search method. The C5.0 function in R supports different tree-based methods; it is, for example, a single tree if ‘trials’ is set to one, or a boosting, analogous to AdaBoost with some tweaks, otherwise. The model can be further improved by considering whether to use a rule-based tree or a classification tree and whether to filter unimportant variables in advance or not. The aforementioned options were all determined using the random search method. Performance indicators of the model were evaluated in the test set using accuracy, specificity, and sensitivity. A seed number was used prior to the data splitting and model training to obtain reproducible results. A receiver operating characteristic (ROC) curve of the performance of the optimal RF model on the test set was visualized using pROC package version 1.10.0 [30]. Visualization of the top 10 important genes of the optimal RF model was performed using GraphPad Prism 6 (GraphPad Software Inc., San Diego, CA, USA). In addition, a quick assessment of the classification of RA and normal tissues as well as OA and normal tissues was conducted using ArrayMining [31].

### 2.6. Random Forests Classification Model Explanation

We applied local interpretable model-agnostic explanations (R package version 0.3.0) in order to explain the prediction decisions of the optimal RF classifier regarding the most important features [32]. A forward selection algorithm with six features was utilized to select the most relevant features. The labels of two were established to observe the explanation of RA and OA. Other parameters were kept as default.

### 2.7. Gene Expression Meta-Analysis

Gene expression meta-analyses between RA and OA, RA and normal tissues, and OA and normal tissues were performed by combined effects size. The statistical heterogeneity was first estimated by Cochran’s Q test. A fixed effects model was used when estimated Q values have approximately followed a chi-square distribution; otherwise, a random effects model was applied. The analysis was conducted in accordance with the published protocol of Networkanalyst [20].

### 2.8. Pathway Enrichment Analysis and Protein Association Network

Gene Ontology and gene set enrichment analysis was conducted using the STRING database with default settings [33]. All differentially expressed genes that had combined effects size greater than 1.5 were input to the software for the enrichment analyses using GO biological processes and Kyoto Encyclopedia of Genes and Genomes annotation. The protein–protein interaction network derived from the STRING knowledgebase was also visualized.

### 2.9. Statistical Significance

Where applicable, a *p*-value of 0.05 or a false discovery rate FDR of 0.05 was used for null-hypothesis testing. The R statistic version 3.4.2 was used as the environment for data processing, machine learning classification and visualization unless specifically indicated [34].

### 2.10. Ethics and Data Availability

Ethical application has been waived by Seoul National University Institutional Review Board (SNU 17-08-047). The datasets analyzed during the current study are available in the GEO repository. GSE1919, GSE39340, GSE36700, GSE55457, GSE55584, GSE55235.

## 3. Results

### 3.1. Variable Importance Measurement and Selection

Error rate-based (ER)-RF variable importance measurement (VIM) has been recommended for feature selection in high-dimensional data [35]. Nevertheless, a previous investigation demonstrated that area under the curve (AUC)-RF VIM outperforms ER-RF VIM, especially in unbalanced class problems [36]. Hence, we applied AUC-RF for variable importance measurement to estimate the score of every gene with the corresponding conditions. The workflow of variable selection is shown in Figure 1a. The measurement was conducted using cross-study normalized data set A derived from three different data sets: GSE1919, GSE39340, and GSE36700 (Appendix A). These available data sets came from various populations (one Asian and two European) and several microarray platforms (Affymetrix U95A, Illumina V 4.0, and Affymetrix 133 Plus 2.0) with a total number of 35 samples were used for variable importance measurement. A larger set of data was used for modeling and testing (described below). The batch effects removed data sets were visualized using principal component analysis (PCA) (Appendix A). As a result, 381 genes exhibited an importance score to the classification of RA and OA greater than zero. Because the package does not provide a handy criterion for selecting the candidates, we finally selected 16 genes with a score of 1.5 × 10^−3^ or higher as potential variables for the classification of RA and OA in unseen cross-study normalized data set B (Figure 1b). The selection was arbitrary to some extent. However, we intended to select the smallest genetic signature that could effectively discriminate RA from OA. A redundant list of genetic signatures—e.g., hundreds of genes—would make it unimaginable for finding the rationales of machine learning-based classification as well as clinical interpretation. Detailed information about the proposed genetic biomarkers can be found in Table 1.

### 3.2. RA and OA can be Accurately Differentiated Using a 16-Gene Signature

To validate our 16-gene signature, we gathered three data sets having a roughly double number of samples that came from the same study in set B (GSE55457, GSE55584, and GSE55235) [9]. It is of importance to note that the authors of the previous study built their own model and obtained good performance in these data sets, and we aimed to show that our signature, derived from a powerful statistical learning method, could also achieve comparative results. In set B, descriptive statistics and univariate data exploratory analysis using the Wilcoxon rank-sum test were performed. There were 59 samples in total, of which OA and RA accounted for 26 samples (44.1%) and 33 samples (55.9%), respectively. The gene expression of all predictors was significantly different (false discovery rate (FDR) < 0.05). The mean, median, and other values as well as the violin plots of gene expression for the 16 predictors can be found in Data S1. Since univariate analysis could not assess the correlations among predictors, PCA was then conducted to obtain better insights into the data. Our signature was actually well adapted to the two merged data sets as shown in PCA analysis. The first PC explained more than 50% variance of the data, and five major PCs cumulatively explained up to 81.9% of the variance in the data. The two classes were relatively well separated in 2D score plot PCA (PC1 + PC2 = 62.3%). Variables having a strong positive or negative effect on PC1 can be considered essential for classification. The three attributes with the highest absolute values were *TMOD1*, *CLU*, and *MXRA7* among the positive factors and *GZMK*, *GZMB*, and *CD8A* among negative factors. Moreover, as will be shown later in this section, *CLU* and *MXRA7* were characterized as important covariates in prediction. Despite these clear findings, however, there was an overlapping region between the two groups, which might make the classification of those cases greatly complicated (Figure 2a). Most of the samples in the two groups distributed within 95% confidence regions for each group, but there were three potential outliers, one from RA and two from OA. Heatmap analysis was also conducted without re-organization to observe the contrasting appearance between two groups (Figure 2b).

There were two relatively clear expression patterns in the heatmap. The gene expression levels of *ALOX5*, *TNFRSF4*, *GZMB*, *IL32*, *KLRD1*, *GZMK*, and *CD8A* were generally higher in RA than in OA. In contrast, the gene expression levels of *RAB22A*, *PTPN3*, *FBXL7*, *MXRA7*, *POP7*, *ANK3*, *SGCA*, *TMOD1*, and *CLU* were generally higher in OA than in RA. For class assignment analysis, data set B was split into training (60%) and test (40%) sets. Data partition was conducted and the balance of class distribution remained between two comparative groups. The training process to estimate the optimal RF model was conducted using grid search; the mtry parameter ranged from one to 16, and the 10-fold cross-validation 5-time repeats method was used. The area under the receiver operating characteristic curve (AUCROC) was used to determine the optimal model. Eventually, the model that had an mtry of 12 exhibited an excellent performance (AUCROC = 1.00) in the training set. Thereafter, we applied the optimal RF model to the external test set. The performance of the RF model was excellent; only one sample was misclassified (Figure 3a). RF variable importance measurement revealed that *IL32*, *SGCA*, *MXRA7*, and *CLU* were the four most important features of the prediction model (Figure 3b).

Local interpretable model-agnostic explanations (LIME) was applied in three representative cases (one correctly predicted OA sample, one correctly predicted RA sample, and one OA sample that was mistakenly predicted as RA) to demonstrate the rules that were applied to classify new observations in the test set of the established model. The first case was confidently predicted to be OA (probability of 0.900) with the following rules regarding gene expression levels: 8.44 < *IL32* ≤ 8.74, 12.20 < *MXRA7* ≤ 12.40, 7.08 < *SGCA* ≤ 7.64, 13.00 < *CLU*, *GZMB* ≤ 6.44, and 7.82 < *PTPN3* ≤ 7.92 (Figure 4a). The second case, on the other hand, was predicted with certainty to be RA (probability of 0.998) with the following rules: 9.34 < *IL32*, *SGCA* ≤ 6.76, *MXRA7* ≤ 12.00, 12.10 < *CLU* ≤ 12.50, 6.77 < *GZMB* ≤ 7.58, *POP7* ≤ 8.62 (Figure 4b). From the two typical cases, we show that the higher expression of *IL32* and *GZMB* as well as the lower expression of *SGCA*, *MXRA*, and *CLU* were preferred for the classification of RA, and vice versa. In addition, the low expression of POP7 was a minor supportive factor for the indication of RA. However, the contributions of minor supportive or contradictory factors were negligible since the roles of top four major factors were prominent in all cases. It is important to mention again that *IL32* and *GZMB* expression levels were generally higher while *MXRA*, *SGCA*, and *CLU* expression levels were generally lower in RA than in OA (Figure 2b). The third example had high expression of *IL32* (9.14) and low expression of *MXRA7* (11.60). The misclassification of this case from OA to RA likely occurred because the expression of *IL32* of the sample was higher than the upper limit of *IL32* in OA (8.74) and the expression of *MXRA7* was lower than the lower limit of *MXRA7* in OA (12.2). This violation was too severe because *IL32* and *MXRA7* were two of the top three most important features in the prediction model. Hence, the high expression level of *SGCA* and *CLU* as well as the low expression level of *RAB22A*, which supported the indication of OA, were not enough for the classifier to correctly differentiate the sample (Figure 4c). The classification of other samples in the test set could be explained using a similar approach as described.

A genetic signature may be more clinically relevant if it can be applied by different machine learning classification methods. Therefore, we introduced our 16-gene signature to four other commonly used classifiers, kNN, naïve Bayes, C5.0, and SVM, to evaluate the robustness and flexibility of the signature. Every model was trained using 10-fold cross-validation with 5-time repeats and AUCROC was used to determine the optimal model. In the kNN model, the data were treated by centering and scaling. The training model with a k value of nine was detected to be optimal (AUCROC = 0.96). The performance in the test set was also excellent (accuracy = 0.96, sensitivity = 0.92, specificity = 1.00). The naïve Bayes model without kernel achieved the best performance (AUCROC = 0.98) which was similar to that of the kNN model in the test set (accuracy = 0.96, sensitivity = 0.92, specificity = 1.00). The training process of the SVM with polynomial kernel produced the optimal model (AUCROC = 0.98) with the following tuned parameters: degree = 3, γ = 0.77, C = 0.16. Likewise, this model exhibited excellent performance in the test set (accuracy = 0.96, sensitivity = 1.00, specificity = 0.90). Finally, the C5.0 model was optimized and validated. In the training set, the model with the following parameters was optimal (AUCROC = 0.93): model = tree, winnow = false, trials = 13. In the test set, it had the worst performance among the tested classifiers although the accuracy was higher than 0.90 (accuracy = 0.91, sensitivity = 1.00, specificity = 0.80). Collectively, we achieved similarly outstanding performance regarding the differentiating of RA from OA to the rule-based method of Woetzel [9]. Confusion matrices of the abovementioned classifiers on the test sets are shown in Figure 5a.

For a quick investigation to evaluate the possibilities of differentiating RA from normal synovial tissues and OA from normal synovial tissues, we applied the class assignment analysis module using ArrayMining [31]. The training and test sets were defined using a 60:40 ratios. We achieved encouraging accuracies, sensitivities, and specificities on the test sets, as shown in Figure 5b. However, the sample sizes of both experiments were small, and the results should be validated by future studies.

### 3.3. Meta-Analysis of Gene Expression and Functional Analysis

To seek the differences in biological processes of RA, OA, and normalcy, we further conducted meta-analysis of all available synovial tissue gene expression data followed by pathway enrichment analysis. Data from synovial tissues of RA and OA, RA and normalcy, and OA and normalcy were introduced to the gene expression meta-analysis to find the differentially expressed (DE) genes with high confidence. Cochran’s Q test suggested the statistical heterogeneity among the input data sets; thus, a random effects model was finally selected for the analysis.

In RA versus OA, 1026 genes were significantly upregulated, and 1421 DE genes were significantly downregulated in RA compared to OA. Among them, 489 DE genes showed a combined effects size of at least 1.5 or higher (236 genes with a combined effect size (cES) of 1.5 or higher in the upregulated group and 253 genes with a cES of −1.5 or lower in the downregulated group). The three most significantly upregulated genes (*CD3D*, *AIM2*, and *IL2RG*) in RA are related to the immune processes. For instance, *CD3D* is one of the genes involved in T cell activation and signaling. Other associated genes, such as *CD3G*, *CD8B*, and *LCK*, were also found to be highly upregulated [37]. Pathway enrichment analysis was conducted separately using the DE genes that had a cES of 1.5 or higher for upregulated genes and of −1.5 of lower for downregulated genes in RA compared to OA. Appendix A shows the two protein–protein interaction networks of RA-OA for upregulated genes and downregulated genes, respectively. Particularly, the major enriched Gene Ontology (GO) biological processes in upregulated genes belonged to various immune responses, including the activation of NF-kB. Similarly, the Kyoto Encyclopedia of Genes and Genomes (KEGG) enriched pathways include, but not limited to, natural killer cell mediated cytotoxicity, cytokine-cytokine receptor interaction, chemokine signaling pathway, and T cell receptor signaling pathway, among others. On the other hand, the enriched GO terms of the downregulated genes reflect the downregulation of cellular development and differentiation. However, there were only two enriched KEGG pathways: glycerolipid metabolism and transcriptional misregulation in cancer. Significantly enriched pathways from the GO and KEGG pathway enrichment analysis are summarized in Table 2 and Appendix A.

In RA versus normal tissues, 973 genes were significantly upregulated, and 690 DE genes were significantly downregulated in RA compared to normal synovial tissues. Among them, 539 genes showed a combined effects size of at least 1.5 or higher (383 genes with a cES of at least 1.5 in the upregulated group and 156 genes with a cES of −1.5 or lower in the downregulated group). The top three enriched GO terms among the upregulated genes were immune response (GO.0006955), immune system process (GO.0002376), and defense response (GO.0006952), while the top three enriched GO terms among the downregulated genes were regulation of protein metabolic process (GO.0051246), cellular response to organic substance (GO.0071310), and positive regulation of multicellular organismal process (GO.0051240). Similarly, 879 DE genes were upregulated, and 524 DE genes were downregulated in OA compared to normal synovial tissues. Among them, 412 DE genes showed a combined effects size of at least 1.5 or higher (274 genes with a cES of at least 1.5 in the upregulated group and 138 genes with a cES of −1.5 or lower in the downregulated group). The top three enriched GO terms among the upregulated genes were response to stress (GO.0006950), death (GO.0016265), and regulation of biological quality (GO.0065008), while the top three enriched GO terms among the downregulated genes were positive regulation of metabolic process (GO0009893), positive regulation of macromolecule metabolic process (GO.0010604), and positive regulation of cellular metabolic process (GO.0031325). The protein–protein interaction networks of RA-Normalcy and OA-Normalcy were simpler than that of RA-OA. The enriched pathways from GO and KEGG analysis of these two networks can be found in Appendix A.

## 4. Discussion

There are standard approaches for the differential diagnosis of RA and OA. Nevertheless, their performance may not be satisfactory in some particular cases. For instance, anti-citrullinated protein antibodies (ACPAs) have been applied for early diagnosis of RA. Nevertheless, the test using ACPAs is only positive in approximately 50% of patients [38]. Other biomarkers, therefore, should be developed to assist the diagnosis in ACPA-negative patients. Recently, serum connective tissue growth factor was reported as a potential diagnostic biomarker for RA [39]. In addition, 99mTc-3PRGD2 scintigraphy has been recently suggested for the early detection of RA in rats and humans [40]. It is also of importance to note that although blood-based tests are prominently used, the diagnostic value of synovial biopsy has been acknowledged [41,42]. Thus, it could help assist the diagnosis RA, OA, as well as other arthritis conditions. However, the potential of the transcriptomic signature of synovial tissue in differentiating RA and OA remains to be explored. High-throughput gene expression has been recognized as an important feature to facilitate personalized medicine [43]. High-throughput gene expression has been commonly used to gain deeper insights into the cellular processes of a specific biological system [44]. Indeed, gene expression profiling has been contributing to the advancement of many fields, including the management of cancers and other immunological disorders [45]. Specifically, the genome-wide alterations in gene expression in RA and OA can be detected in advance [3,46]. Recently, a method using a transcriptome-based rule set was developed to differentiate RA from OA and normal controls; the overall assessment parameters of classification were approximately 90% or higher [9]. Similarly, a 12-gene signature derived from CD4^+^ T cells of early inflammatory arthritis patients has been shown to accurately predict subsequent development of RA from non-RA among undifferentiated arthritis with a sensitivity of 68% and specificity of 70% [47]. These results are encouraging and worth validating on a larger scale to assess their reproducibility. Moreover, a wide range of supervised machine learning techniques can be employed toward the final goal, which is to classify a particular sample. However, small sample sizes and noisy data of the microarray-based high-throughput gene expression have become limitations since the generalization of the results is not guaranteed and the cross-study validation often failed to replicate the results [48]. Potential solutions, such as cross-platform normalization or cross-study normalization for multi-data integration, have been introduced and have achieved considerable success [49,50]. In addition, conducting feature selection prior to classification can help reduce highly noisy attributes while simultaneously improving the prediction accuracy for the class assignment analysis [51,52]. Finally, but importantly, the current limitations in the progress of diagnosing and determining the prognosis of RA may come from the substantial portion of mechanism-based elucidation. An unbiased, data-driven approach emphasizing the roles of a small subset of genes that are distinctly different between comparative conditions may additionally provide potential solutions for early diagnosis. In our study, 16 genes were selected as the biomarker signature for differentiating between RA and OA using an unbiased, data-driven method. The validity of the proposed signature was then challenged using independent data sets with various machine learning classification algorithms. The classification performances derived from five commonly used classifiers in the external test set suggested that RA and OA could be accurately differentiated with an accuracy higher than 90% using our 16-gene panel and machine learning approaches, even by ‘weak’ classifiers such as kNN. Accordingly, these genes are worth considering as novel biomarkers for the early differential diagnosis of RA and OA. Among others—*CD8A*, *GZMB*, and *KLRD1*, for instance—are strongly associated with one another in the STRING protein–protein interaction network and are associated with lymphocyte-mediated immunity.

Although the 16-gene panel was derived using a data-driven approach, several individual biomarkers showed a clear biological connection with the arthritis conditions (Appendix A). However, the role of *TMOD1*, *SGCA*, *RAB22A*, *ANK3*, *PTPN3*, *FBXL7*, *CLU*, *POP7*, and *MXRA7* in inflammatory diseases, especially in arthritis, has not been studied thoroughly. Further studies are needed to clarify their biological function. Moreover, the recurrent inflammation of the synovial tissues suggests the complexity of the immune responses in the pathobiology of RA, as also being mentioned in a recent meta-analysis [53]. Many other biological pathways are involved in various heterogeneous processes at different stages of the disease [54]. Unlike other forms of arthritis, destruction is the nature of RA. Hence, early diagnosis and management are especially important. Pain, structural damage, functional loss, comorbidity, and other complications of RA lead to the permanent disability of patients as well as to increased socioeconomic cost [55]. On the other hand, the main pathological condition of OA is the degradation of cartilage, which eventually results in joint dysfunction [56]. Our functional analysis showed general agreement; thus, confirmed the previously reported differences in the mechanisms of the two diseases and suggested novel pathways that are prominently enriched in RA compared to OA [57,58,59]. From this data, further mechanistic studies are needed to obtain better insights into the pathophysiology of RA as well as other forms of arthritis in order to improve patient management.

Our work has some limitations that should be explicitly stated. All analyses were conducted using tissues that might not represent the best approach for RA and OA differential diagnosis. However, synovial tissues are usually informatively rich with respect to the underlying pathological processes of the diseases; therefore, our findings may give additional information for the improvement of the patient management and mechanistic studies. Second, only transcriptome information of patients was utilized for the differential analysis so the effects of confounding factors remain unknown. Additionally, the lack of assessment using synovial fluids or serology prevented us from conducting deeper data mining. However, we demonstrated that sophisticated algorithms are powerful in introducing and validating potential biomarker candidates. We also illustrated an application of a human-friendly rule set to interpret the ‘black-box’ RF model. Further investigations that take into account the patients’ bio-parameters, omics data, and statistical learning methods will greatly improve our understanding and clinical practice to provide better care for the patients.

## 5. Conclusions

The differential diagnosis of RA from other forms of arthritis, especially OA, using synovial biopsy is a novel approach. Our study introduced and successfully validated a 16-gene signature derived from gene expression profiling data of synovial tissues that could be employed for the classification of RA and OA. The signature may also help more specifically treat patients with RA condition. The novelty of this investigation is based on the utility of powerful statistical learning methods for variable selection and modeling. Moreover, we applied an explainer algorithm to interpret the rationale of the black-box predicting algorithms. Further clinical studies are warranted to validate this biomarker for the diagnosis of rheumatoid arthritis patients.

## Figures and Tables

**Figure 1 jcm-08-00050-f001:**
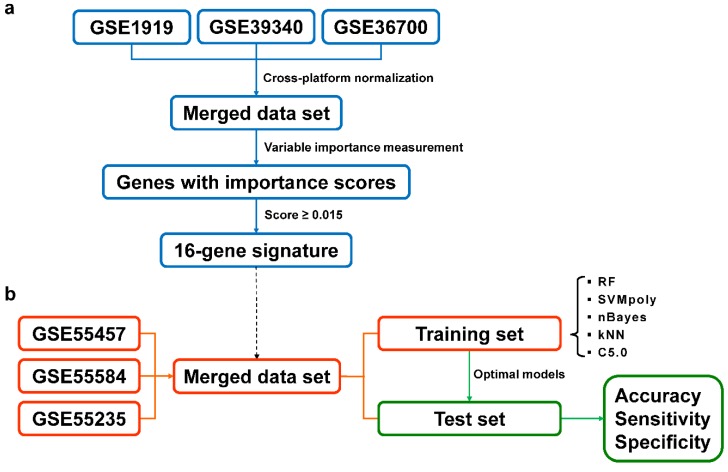
Workflow of variable selection and machine learning classification. (**a**) Variable selection workflow. (**b**) Classification analysis workflow. RF: random forests; SVM: support vector machines; kNN: k nearest neighbors. GSE55457, GSE55584, and GSE55235 are from the same study and were used for modeling and validation of the data-driven proposed biomarkers.

**Figure 2 jcm-08-00050-f002:**
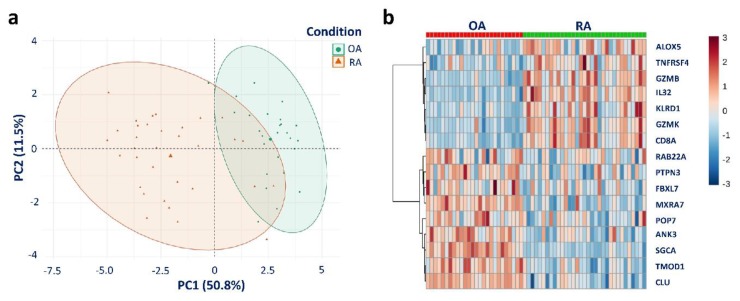
PCA and heatmap analysis of the data set with the 16-gene signature. (**a**) The sum of the two principal component is 62.3%. (**b**) Seven genes are upregulated, and nine genes are downregulated in RA compared to OA. PCA: Principal component analysis; RA: Rheumatoid arthritis; OA: osteoarthritis.

**Figure 3 jcm-08-00050-f003:**
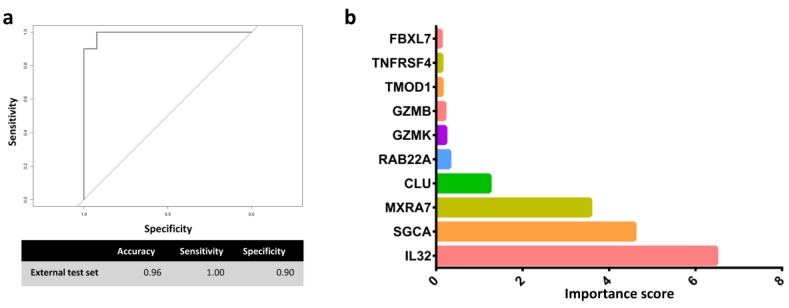
Properties of the random forests model. (**a**) The ROC curve, accuracy, sensitivity, and specificity of the external the test set of the optimal random forest model. (**b**) The top 10 most important features of the optimal random forests model on the training set.

**Figure 4 jcm-08-00050-f004:**
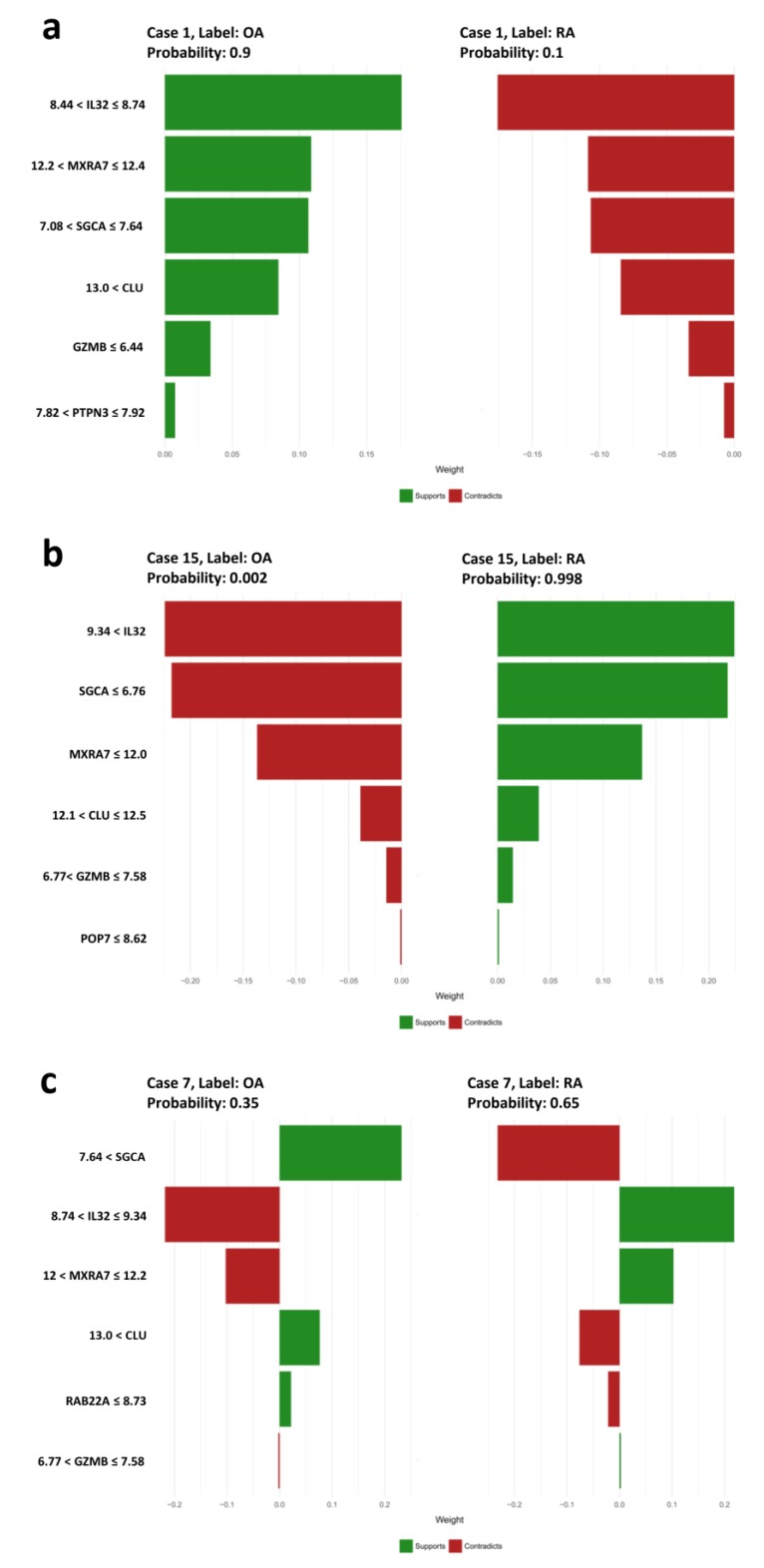
The use of LIME to explain the model′s predictions. (**a**) Corrected classification of an OA sample. (**b**) Corrected classification of an RA sample. (**c**) Noncorrected classification of a sample from OA to RA. RA: Rheumatoid arthritis; OA: osteoarthritis.

**Figure 5 jcm-08-00050-f005:**
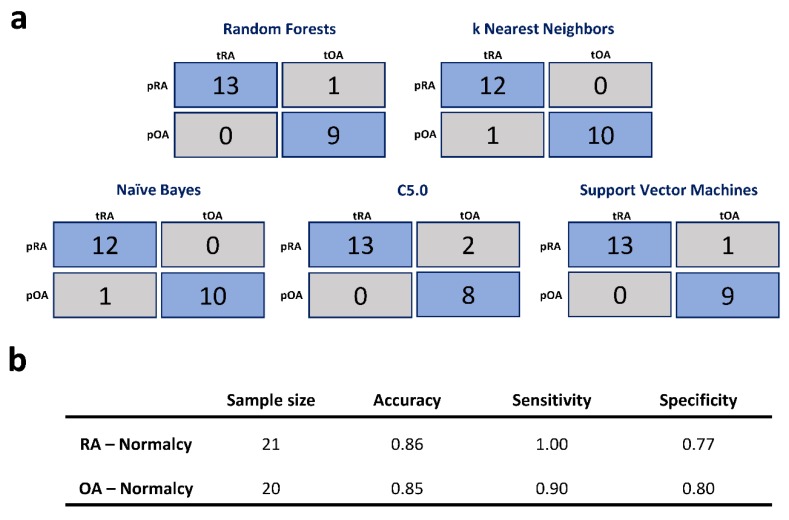
Summarization of the classification analyses. (**a**) Confusion matrices of the five classifiers on the test set of RA versus OA. (**b**) Prediction performances on the test sets of RA versus normalcy and OA versus normalcy. RA: Rheumatoid arthritis; OA: osteoarthritis.

**Table 1 jcm-08-00050-t001:** The 16-gene signature derived from AUC-RF based variable selection.

Entry ID	Approved Symbol	Approved Name	Chromosomal Location	Combined Effects Size ^1^
7111	*TMOD1*	Tropomodulin 1	9q22.33	−2.16
10248	*POP7*	POP7 homolog, ribonuclease P/MRP subunit	7q22.1	−0.87
6442	*SGCA*	Sarcoglycan alpha	17q21.33	−2.74
3824	*KLRD1*	Killer cell lectin like receptor D1	12p13	1.35
240	*ALOX5*	Arachidonate 5-lipoxygenase	10q11.21	1.36
57403	*RAB22A*	RAB22A, member RAS oncogene family	20q13.32	−1.20
288	*ANK3*	Ankyrin 3	10q21.2	−1.92
5774	*PTPN3*	Protein tyrosine phosphatase, non-receptor type 3	9q31	−1.31
3003	*GZMK*	Granzyme K	5q11.2	2.92
1191	*CLU*	Clusterin	8p21.1	−2.17
3002	*GZMB*	Granzyme B	14q12	2.61
23194	*FBXL7*	F-box and leucine rich repeat protein 7	5p15.1	−0.93
7293	*TNFRSF4*	TNF receptor superfamily member 4	1p36.33	1.26
9235	*IL32*	Interleukin 32	16p13.3	2.73
439921	*MXRA7*	Matrix remodeling associated 7	17q25.1	−2.19
925	*CD8A*	CD8a molecule	2p11.2	2.83

^1^: Adopted from the the section of functional meta-analysis, which was independent to the biomarker selection.

**Table 2 jcm-08-00050-t002:** Representative Gene Ontology (GO) biological processes and Kyoto Encyclopedia of Genes and Genomes (KEGG) enriched pathways of differentially expressed (DE) genes between RA and OA.

ID	Annotation	False Discovery Rate	RA versus OA
GO.0006955	Immune response	8.86 × 10^−50^	Upregulation
GO.0050776	Regulation of immune response	1.44 × 10^−41^	Upregulation
GO.0002376	Immune system process	2.64 × 10^−41^	Upregulation
GO.0006952	Defense response	6.94 × 10^−39^	Upregulation
GO.0002684	Positive regulation of immune system process	4.52 × 10^−36^	Upregulation
GO.0048731	System development	8.90 × 10^−12^	Downregulation
GO.0007275	Multicellular organismal development	3.68 × 10^−11^	Downregulation
GO.0044767	Single-organism developmental process	2.74 × 10^−9^	Downregulation
GO.0048856	Anatomical structure development	1.29 × 10^−8^	Downregulation
GO.0051239	Regulation of multicellular organismal process	5.14 × 10^−8^	Downregulation
KEGG.4650	Natural killer cell mediated cytotoxicity	1.27 × 10^−16^	Upregulation
KEGG.5340	Primary immunodeficiency	1.27 × 10^−16^	Upregulation
KEGG.4060	Cytokine-cytokine receptor interaction	3.14 × 10^−16^	Upregulation
KEGG.4064	NF-kappa B signaling pathway	1.26 × 10^−14^	Upregulation
KEGG.4062	Chemokine signaling pathway	1.33 × 10^−11^	Upregulation
KEGG.561	Glycerolipid metabolism	0.0142	Downregulation
KEGG.5202	Transcriptional misregulation in cancer	0.0276	Downregulation

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
