# Peer review of "Efficacy of Integrating a Novel 16-Gene Biomarker Panel and Intelligence Classifiers for Differential Diagnosis of Rheumatoid Arthritis and Osteoarthritis"

_jcm, 2019, doi:10.3390/jcm8010050_

Round 1
Reviewer 1 Report
Many thanks for asking for comments on this manuscript
Major comments:
The authors seem to equate OA and RA in the introduction , this could be clarified and the prevalence of each should be clarified.
The use of the statistical methods is not clear and does not appear to be consistent with published data. i would suggest a sensitivity analysis using more standard statistics.
Author Response
Reviewer #1:
Overall comment:
Many thanks for asking for comments on this manuscript
Answer:
We really appreciate the reviewer for your time and effort in evaluating our works. The manuscript is amended and the details can be seen below.
Specific comments:
Comment:
The authors seem to equate OA and RA in the introduction, this could be clarified and the prevalence of each should be clarified.
Answer:
We appreciate your careful review. We totally agree with the reviewer that the prevalence information is important. Thus, we added the following aspects as you suggested in this comment to clarify the prevalence of each disease (line 41-44): “Depending on the case definition and joint sites under study, the prevalence of RA was at 0.5–1.1% while that of OA was much more common, ranging from 5% of the hip and 33% of the knee to 60% of the hands in adults 65 years of age or older.”
Comment:
The use of the statistical methods is not clear and does not appear to be consistent with published data. i would suggest a sensitivity analysis using more standard statistics.
Answer:
We would like to thank the reviewer for indicating this aspect. We considered your opinion seriously and rechecked the applied methods accordingly. Additionally, we arranged a meeting with the statistical experts of our college including two authors of the current works (Seongoh Park and Johan Lim). Major information is mentioned below:
Data preparation and preprocessing
We applied the standard workflow of data preprocessing (by using R package from Bioconductor) after downloading from GEO. There are some example papers from our research group that applied the similar strategies and accepted by the community (doi: 10.1371/journal.pone.0148818, 10.18632/oncotarget.22689).
The batch effects removal during cross-platform normalization has been discussed. We applied ComBat method, one of the most well-known and tested techniques among others (doi: 10.3390/microarrays4030389).
Classification aspect:
We applied the standard approach for this mathematical problem, as guided by the author of the Caret package himself and the book Applied Predictive Modeling, ISBN 978-1-4614-6849-3). Of note, we took the randomness effect seriously so several random seeds have been tested that confirmed the robustness of our analysis. Certainly, there might be some minor differences with respect to the seeds, version of software, and applied packages. However, our main results hold true.
During the revision, the logistic regression, as one of conventional and standard approaches in the classification task, is also performed to provide a good benchmark along with methods we used. Though as simple as nothing to be tuned, the model shows satisfactory classification performance thanks to carefully selected biomarkers in the training as well as the test set. More specifically, only 3.5 (in average through 20 repetitions) of 23 test samples are misclassified from RA to OA, or vice versa. The value of data-driven strategy has been acknowledging and are developing quickly (doi: 10.1093/bioinformatics/bty825). Please understand that we strictly follow this approach in the current study.
Other aspects:
The Wilcoxon rank-sum test is a standard method for univariate analysis between two independent groups.
The horizontal genomics meta-analysis was conducted using NetworkAnalyst. We followed strictly the protocol provided by the authors, which is also published in Nature Protocol (doi: nprot.2015.052).
Collectively, we believe that all other applied statistical methods used in the current work are suitable. Please advise us if you think in other respects. Thank you again for your valuable suggestion that gave us a chance to recheck our work systematically. We hope that our explanation and declaration are satisfactory. Otherwise, please kindly advise us again.
Reviewer 2 Report
Authors aim to search for a novel data-driven gene signature of synovial tissues to differentiate RA from OA patients by using microarray-based transcriptome samples. Authors use intelligence classifiers to detect a gene signature that can effectively differentiate rheumatoid arthritis (RA) from osteoarthritis (OA).
The work is interesting, but the background does not fit the aim of the study. In particular, the pathogenic mechanisms and clinical presentation of RA and OA differ greatly, due to the inflammatory and degenerative nature of the two diseases respectively. For example the age of onset, the laboratory tests and the biomarkers (rheumatoid factors and anti-citrullinated peptide antibodies) are very different in the two conditions. Therefore, sentences such as in row 348 in the Discussion section "there are currently no optimal criteria for the differential diagnosis of RA and OA..." should be avoided. Differential diagnosis is helpful in a subset of elder patients with inflammatory presentation of osteoarthritis or in late-onset RA patients seronegative for typical disease biomarkers. Therefore, their work is justified for research purpose and for a limited subset of patients and this should be stated more clearly in the introduction.
-Being a research article, the manuscript should be subdivided in the suggested Research manuscript sections: Introduction, Materials and Methods, Results, Discussion, Conclusions (optional).
-the Discussion is too long. It is interesting that Authors describe the function or supposed fuction of each gene, but a table may well summarise these data.
-limitations and strengths of the study should be highlighted.
-the sentence on row 449 should be removed as it does not fit in the discussion.
-English needs revision for some typos.
Author Response
Overall comment:
Authors aim to search for a novel data-driven gene signature of synovial tissues to differentiate RA from OA patients by using microarray-based transcriptome samples. Authors use intelligence classifiers to detect a gene signature that can effectively differentiate rheumatoid arthritis (RA) from osteoarthritis (OA).
Answer
Please accept our appreciation for your time and effort on helping us improve the current work. We read carefully all comments and carefully revised our manuscript. All raised concerns are addressed with great care. The manuscript was amended and the details can be seen below.
Specific comments:
Comment:
The work is interesting, but the background does not fit the aim of the study. In particular, the pathogenic mechanisms and clinical presentation of RA and OA differ greatly, due to the inflammatory and degenerative nature of the two diseases respectively. For example the age of onset, the laboratory tests and the biomarkers (rheumatoid factors and anti-citrullinated peptide antibodies) are very different in the two conditions. Therefore, sentences such as in row 348 in the Discussion section "there are currently no optimal criteria for the differential diagnosis of RA and OA..." should be avoided. Differential diagnosis is helpful in a subset of elder patients with inflammatory presentation of osteoarthritis or in late-onset RA patients seronegative for typical disease biomarkers. Therefore, their work is justified for research purpose and for a limited subset of patients and this should be stated more clearly in the introduction.
Answer
We appreciate the reviewer for mentioning the concern on this important feature. We totally agree with your opinion. Therefore, we have made the corrections accordingly.
In line 44-47, we mentioned the pathogenic mechanisms of RA and OA, emphasized particularly on the early stage: “RA is a chronic autoimmune disease that exhibits persistent synovial and systematic inflammation along with the existence of autoantibodies. On the other hand, OA has been characterized as a non-inflammatory degenerative joint disease although synovial inflammation is a debatably important feature. OA and RA are pathophysiologically different but share similar and overlapping features in terms of underlying mechanisms.”
Particularly, the following sentence was added to the introduction (line 67-69): “The results from this investigation are expected to be helpful in some particular populations, such as elder patients with inflammatory presentation of OA and patients without typical biomarkers.”.
In line 72-74, we mentioned that our approach is for “assisting” the blood-based tests in complicated cases: “In conclusion, our findings may assist the blood-based laboratory tests to improve the accuracy of diagnosis and successful rate of clinical interventions.”
We hope that provided information is satisfactory to represent the aim of the current work.
We also rewrote the sentence in line 358-359 according to your comment: “There are standard approaches for the differential diagnosis of RA and OA. Nevertheless, their performance may not be satisfactory in some particular cases.”
Comment:
-Being a research article, the manuscript should be subdivided in the suggested Research manuscript sections: Introduction, Materials and Methods, Results, Discussion, Conclusions (optional).
Answer
We thank the reviewer for the kind reminder. We amended the structure of the manuscript based on the instructions of the journal. The amended structure follows:
Front matter: Title, Author list, Affiliations, Abstract, Keywords
Research manuscript sections: Introduction, Materials and Methods (line 75), Results, Discussion, Conclusions (optional).
Back matter: Supplementary Materials, Acknowledgments, Author Contributions, Conflicts of Interest, References.
Comment:
-the Discussion is too long. It is interesting that Authors describe the function or supposed fuction of each gene, but a table may well summarise these data.
Answer
We really appreciate your comment on this point. We have shortened the discussion section as request to improve the quality and readability of the manuscript. In addition, the functions or supposed functions of each biomarker are now can be found in the supplementary file. Importantly, we created Table S7 that contains all biological and supposed biological functions of individual markers (line 403). Thus, the interested readers will find all information they need. Thank you very much for your valuable comment.
Comment:
-limitations and strengths of the study should be highlighted.
Answer
We appreciate the reviewer for mentioning the concern on this important feature. We added the limitations (and advantages) paragraph at the end of the discussion section. Following sentences were added:
Line 418-429: “Our work has some limitations that should be explicitly stated. All analyses were conducted using tissues that might not represent the best approach for RA and OA differential diagnosis. However, synovial tissues are usually informatively rich with respect to the underlying pathological processes of the diseases; therefore, our findings may give additional information for the improvement of the patient management and mechanistic studies. Second, only transcriptome information of patients was utilized for the differential analysis so the effects of confounding factors remain unknown. Additionally, the lack of assessment using synovial fluids or serology prevented us from conducting deeper data mining. However, we demonstrated that sophisticated algorithms are powerful in introducing and validating potential biomarker candidates. We also illustrated an application of a human-friendly rule set to interpret the "black-box" RF model. Further investigations that take into account the patients' bio-parameters, omics data, and statistical learning methods will greatly improve our understanding and clinical practice to provide better care for the patients.”
Comment:
-the sentence on row 449 should be removed as it does not fit in the discussion.
Answer
Thank you very much for your careful evaluation. We removed the sentence to maintain the logical flow of the discussion.
Comment:
-English needs revision for some typos.
Answer
Please accept our apologies for the shortcomings during the production of the manuscript. All authors have read and critically revised the whole manuscript.
Reviewer 3 Report
It is a novel study and overall is well-written. However, there are some issues need to be clarified. 1) As stated by the Authors sample sizes of both training and test sets are small and the results need to be verified by the future studies.2) The methodology explained in detail but there are no information about disease characteristics like early or established disease, serology etc 3) Did the Authors analyze any synovial fluid or plasma for known RA markers like ACPA, RF? 4) Did the Authors seek for association of their candidates and RA findings like radiology, proof of synovitis, serology?
Author Response
Overall comment:
It is a novel study and overall is well-written. However, there are some issues need to be clarified.
Answer
Thank you very much for your kind advice with regard to the content of our paper. We read all the review comments carefully and tried our best to revise the manuscript accordingly.
Specific comments:
Comment:
1) As stated by the Authors sample sizes of both training and test sets are small and the results need to be verified by the future studies.
Answer
Thank you for your suggestion. We totally agree with this opinion (as also stated in our manuscript). More efforts should be made to validate our findings. Also, further clinical studies are warranted to validate proposed biomarkers, especially synovial tissue, for the diagnosis of rheumatoid arthritis patients. We are also trying to design a long-term following studies and hopefully will get breakthrough outcome.
Comment:
2) The methodology explained in detail but there are no information about disease characteristics like early or established disease, serology etc.
Answer
We appreciate and totally agree with the reviewer at this point. We have tried our best to get all possible information from published data (Table S1). Some information could not be determined explicitly even when look closely into the original works. Thus, we mentioned this fact in the limitation paragraph. Please read the answer of the following comments for more information. However, as also demonstrated by Woetzel et al. (doi: 10.1186/ar4526), it would be difficult (but also great) to use transcriptome information only to detect and conduct differential diagnosis of RA and OA. Certainly, following studies are needed for better assessment of our findings.
Comment:
3) Did the Authors analyze any synovial fluid or plasma for known RA markers like ACPA, RF?
4) Did the Authors seek for association of their candidates and RA findings like radiology, proof of synovitis, serology?
Answer
We thank the reviewer for your careful evaluation. We could not get access to the synovial fluid or plasma as well as radiology, proof of synovitis, and serology. Combined with the above comment and the reviewer 2’s comment, we prepared the limitation paragraph to demonstrate the shortcoming and potential solutions of these problems.
Line 418-429: "Our work has some limitations that should be explicitly stated. All analyses were conducted using tissues that might not represent the best approach for RA and OA differential diagnosis. However, synovial tissues are usually informatively rich with respect to the underlying pathological processes of the diseases; therefore, our findings may give additional information for the improvement of the patient management and mechanistic studies. Second, only transcriptome information of patients was utilized for the differential analysis so the effects of confounding factors remain unknown. Additionally, the lack of assessment using synovial fluids or serology prevented us from conducting deeper data mining. However, we demonstrated that sophisticated algorithms are powerful in introducing and validating potential biomarker candidates. We also illustrated an application of a human-friendly rule set to interpret the "black-box" RF model. Further investigations that take into account the patients' bio-parameters, omics data, and statistical learning methods will greatly improve our understanding and clinical practice to provide better care for the patients."
Round 2
Reviewer 1 Report
I have no further comments
Reviewer 2 Report
Authors have now followed the reviewers' suggestions.
Reviewer 3 Report
No additional comment